# The Influence of Maternal Vitamin E Concentrations in Different Trimesters on Gestational Diabetes and Large-for-Gestational-Age: A Retrospective Study in China

**DOI:** 10.3390/nu14081629

**Published:** 2022-04-14

**Authors:** Qianling Zhou, Mingyuan Jiao, Na Han, Wangxing Yang, Heling Bao, Zhenghong Ren

**Affiliations:** 1Department of Maternal and Child Health, School of Public Health, Peking University, Beijing 100191, China; sibulanliqi@pku.edu.cn (W.Y.); baohl@bjmu.edu.cn (H.B.); rzhong65@126.com (Z.R.); 2Tongzhou Maternal and Child Health Care Hospital of Beijing, Beijing 101101, China; 13522833318@163.com (M.J.); hanna_7656@163.com (N.H.)

**Keywords:** vitamin E, gestational diabetes mellitus (GDM), large-for-gestational-age (LGA)

## Abstract

Vitamin E can protect pregnant women from oxidative stress and further affect pregnancy outcomes. This study aimed to investigate maternal vitamin E concentration in each trimester and its associations with gestational diabetes (GDM) and large-for-gestational-age (LGA). The data were derived from Peking University Retrospective Birth Cohort in Tongzhou, collected from 2015 to 2018 (n = 19,647). Maternal serum vitamin E were measured from blood samples collected in each trimester. Logistic regressions were performed to analyze the association between maternal vitamin E levels and outcomes. The median levels of maternal vitamin E increased from the first (10.00 mg/L) to the third (16.00 mg/L) trimester. Among mothers who had inadequate vitamin E levels, most of them had excessive amounts. Excessive vitamin E level in the second trimester was a risk factor for GDM (aOR = 1.640, 95% CI: 1.316–2.044) and LGA (aOR = 1.334, 95% CI: 1.022–1.742). Maternal vitamin E concentrations in the first and second trimesters were positively associated with GDM (first: aOR = 1.056, 95% CI: 1.038–1.073; second: aOR = 1.062, 95% CI: 1.043–1.082) and LGA (first: aOR = 1.030, 95% CI: 1.009–1.051; second: aOR = 1.040, 95% CI: 1.017–1.064). Avoiding an excess of vitamin E during pregnancy might be an effective measure to reduce GDM and LGA. Studies to explore the potential mechanisms are warranted.

## 1. Introduction

Vitamin E is a lipid-soluble antioxidant that corrects oxidative imbalance and protects tissue from damage [1,2]. For pregnant women, oxidative stress is associated with poor perinatal outcomes. Maternal vitamin E levels during pregnancy have been found to have a significant impact on pregnancy and birth outcomes [3,4]. For instance, maternal plasma concentrations of α-tocopherol at 16 and at 28 weeks of gestation were positively related to fetal growth and associated with an increased risk of delivering large-for-gestational-age (LGA) infants [5]. However, excessive vitamin E may lead to abortion and interfere with fetal development [6]. The majority of the existing evidence examined maternal vitamin E at one or two time points. It is important to detect the association between maternal vitamin E at different trimesters and pregnancy outcomes in order to identify sensitive time period for monitoring and intervention.

Gestational diabetes mellitus (GDM) is one of the most common medical complications in pregnancy and is greatly impacted by maternal nutrition status. GDM is a risk factor for many adverse pregnancy outcomes, such as maternal preeclampsia, stillbirth, fetal intrauterine growth retardation, and macrosomia [7]. GDM affects 2% to 25% of pregnancies globally [8]. According to a recent meta-analysis involving 79,064 Chinese participants from 25 papers, the total incidence of GDM in mainland China was 14.8% (95% confidence interval 12.8–16.7%) [9]. Studies have not yet reached a consensus on the influence of maternal vitamin E concentration on the occurrence of GDM. A meta-analysis found that the level of vitamin E was significantly lower in GDM women compared to healthy pregnant women [10]. However, two recent studies found opposite results and reported that serum vitamin E levels of GDM patients were excessive in mid- and late pregnancy [11,12]. Notably, the majority of previous studies examined vitamin E levels at late pregnancy (after the diagnosis of GDM); therefore, the influence of maternal vitamin E on the occurrence of GDM could not be determined.

Infant birth weight is an important indicator of fetal growth. LGA refers to babies whose birth weight is above the 90th percentile of the average weight of the same gestational age. The global incidence of LGA was 9.4%–18.01% in recent years [13,14,15]. A review of the literature published between 1989 and 2019 found that the prevalence of LGA ranged from 4.3% (Korea) to 22.1% (China) in Asia, while different growth charts were used to define LGA [16]. According to the study “The Chinese Collaborative Study Group for Etiologies of NICU Deaths”, the risk of neonatal death with LGA at an early stage was 1.94 times higher than that of normal newborns (OR = 2.938, 95% CI: 1.346–6.416) [17]. Gadhok et al. [18] found that low levels of vitamin E of pregnant women who admitted in hospital for delivery were associated with fetal intrauterine growth restriction. Maternal vitamin E deficiency was associated with low birth weight [19]. In contrast, a high serum vitamin E concentration in pregnant women could lead to macrosomia [20]. However, according to the current literature, the association between maternal vitamin E and LGA has not been explored.

The existing evidence was limited by its relatively small sample size, the examination of vitamin E in only one or two trimesters, the absence of some important confounding factors, and a lack of details in the study design. The present study was conducted to examine maternal vitamin E concentrations in different trimesters and to explore their associations with adverse pregnancy outcomes, including GDM and LGA. The present study was improved by using a retrospective study design with a relatively large sample size and would demonstrate the importance of adequate vitamin E levels during pregnancy in the prevention of adverse pregnancy and birth outcomes.

## 2. Materials and Methods

### 2.1. Study Design and Setting

The data was derived from Peking University Retrospective Birth Cohort in Tongzhou based on the hospital information system. This study conducted from July 2015 to January 2018 in the Tongzhou Maternal and Child Health Care Hospital of Beijing. Tongzhou, located in southeastern Beijing, is the city’s deputy administrative center. The district emphasizes the development of culture, education, science, and tourism. At the end of 2018, there were a total of 1.58 million residents in Tongzhou district.

### 2.2. Study Population

Participants were pregnant women who had a maternity check-up at the Tongzhou Maternal and Child Health Care Hospital of Beijing between July 2015 and December 2017 (n = 57,332), had their vitamin E measured in any trimester during pregnancy, delivered at the hospital, and whose complete background information was available. Some values that were considered as abnormal were recorded as a missing value in the analyses, including infant birth weight < 354 g, maternal height < 110 cm or > 200 cm, pre-pregnancy weight > 130 kg or < 30 kg, delivery gestational age > 43 weeks, and parity > gravidity. Alongside this, women with stillbirth and multiple pregnancies were excluded from the analyses and women with other pregnancy complications and pregnancy risk factors (i.e., thyroid disease, tumor, hypertensive disorder complicating pregnancy, asthma, ovarian cyst, acute fatty liver of pregnancy, taking drugs contraindicated during pregnancy, etc.) were excluded in the study of GDM. All pregnant women in this study agreed to provide their information to the hospital.

### 2.3. Vitamin E Measurement

Participants had vitamin E measurements in each trimester after the collection of blood samples. The first step was to collect venous blood samples from the participants and store the serum at −80 degrees Celsius in a light-proof place. This was followed by pretreatment with centrifugation and extraction to ensure that impurities and proteins were discarded and that the vitamins were completely separated. The high-performance liquid chromatography method was conducted to measure vitamin E concentrations, in which ɑ-Tocpherol was used as the standard, and 1290 Infinity high-performance liquid chromatography (Agilent) and C18 chromatographic column were applied. A flow rate of 1.0 mL/min was adopted in the measurement. Vitamin E concentrations of the testing and quality control samples were calculated from the standard curve equation, which was established by measuring standard substances. The criterion for determining that a batch was within the control was that the concentrations of the quality control samples were all in the range of mean ± 2 S.D.

### 2.4. Data Collection

Data were obtained from the hospital’s electronic information system. Data used in this analysis included maternal vitamin E concentration in each trimester, participants’ sociodemographic characteristics, parity, maternal pre-pregnancy BMI, gestational weeks at the time of vitamin E tests, folic acid usage (between 1 to 3 months pre-pregnancy and 3 months post-conception), maternal and neonatal outcomes, including GDM, multiple pregnancy, infant birth weight, and preterm birth.

Ethics approval was obtained from the Institutional Review Board (IRB 00001052-19006) of the Peking University Health Science Centre before the implementation of the research. All data used for analysis were anonymous.

### 2.5. Variables and Definitions

The outcome variables of this study included GDM and LGA. GDM was a condition in which women without previously diagnosed diabetes exhibited high blood glucose levels during pregnancy. In the second trimester, all pregnant women routinely took the Oral Glucose Tolerance Test (OGTT) in a morning after eight hours’ fasting. According to the standard set by the International Association of Diabetes and Pregnancy Study Groups, GDM was diagnosed if there was one or more abnormal values for the following: fasting blood glucose level ≥ 5.1 mmol/L, blood glucose level ≥ 10.0 mmol/L one hour after the consumption of glucose, and blood glucose level ≥ 8.5 mmol/L two hours after the consumption of glucose [21]. LGA is defined as the birth weight of infants above the 90 th percentile of the average weight of the same gestational age, according to the international anthropometric standards [22]. Infant birth weight was measured immediately after delivery by midwives.

Serum vitamin E was an independent variable. According to the WHO, vitamin E concentrations were classified into three groups: deficient (<5.0 mg/L), adequate (5.0–20.0 mg/L), and excessive (≥20.0 mg/L) [23]. Weight and height self-reported by the participants in the first antenatal examination were used to calculate pre-pregnancy BMI, which was further divided into four groups (<18.50, 18.50–23.99, 24.00–27.99, ≥28.00 kg/m^2^). Preterm birth referred to live births with <37 gestational weeks [24]. A multiple pregnancy occurred when more than one fetus was delivered in a single pregnancy. The first, second, and third trimesters were indicated as a gestational week of ≤13 weeks, between 14 and 27 weeks, and of ≥28 weeks, respectively.

### 2.6. Statistically Analyses

Vitamin E levels during each trimester were assessed for normality by the Kolmogorov-Smirnov test. The median (IQR) was used to describe the central and dispersion tendency of vitamin E concentrations in each trimester. Categorical variables were described by frequencies and percentages. Differences between outcomes (GDM and LGA) in different maternal characteristics and maternal concentrations were explored by χ^2^ tests, Fisher’s exact tests, or Mann-Whitney U-test. Multivariate logistic regression analyses were further performed to examine the independent effect of vitamin E concentration/status in each trimester on the outcomes. Variables having a *p* < 0.1 in the univariate analyses were adjusted in the multivariate analyses as potential confounders. The association was presented using the odds ratio (OR) and its 95% confidence interval (CI). Data analyses were performed by SPSS version 20, and a *p* < 0.05 was considered statistically significant. In addition, restricted cubic splines (RCS) were performed to explore the dose-response relationships of maternal vitamin E levels and the risks of GDM and LGA, with the use of 20.0 mg/L as the reference value. RCS was performed using R 4.1.3.

## 3. Results

### 3.1. General Characteristics of the Participants

A total of 19,647 women were included in this study. The majority of them were between 21 and 30 years old (64.9%), were primiparous (60.3%), were of Han ethnicity (94.0%), had a high school education level and/or above (78.7%), had a normal pre-pregnancy BMI (63.1%), and had used folic acid between 1–3 months pre-pregnancy and 3 months post-conception (91.2%). In this study population, the prevalence of preterm birth was 3.9% (Table 1).

### 3.2. Maternal Vitamin E Status during Three Trimesters

There were 16,705, 5520 and 2190 women taking vitamin E measurements in the first, second and third trimesters, respectively. Table 2 shows an increasing trend of the mean vitamin E concentrations (10.00 mg/L, 14.60 mg/L, 16.00 mg/L) and the proportion of excessive vitamin E concentrations (0.2%, 9.0%, 13.6%) from the first to the third trimester. However, vitamin E concentrations were adequate for the majority of participants in all trimesters (>86%). Only in the first trimester was there a deficient concentration of vitamin E (0.1%) (Table 2).

### 3.3. The Association between Maternal Vitamin E Status/Concentration and Outcomes (GDM and LGA), Univariate Analyses

Appendix A show the univariate associations between maternal characteristics and outcomes (including GDM and LGA). When looking at the univariate association between maternal vitamin E status/concentration and outcomes, an excessive level of vitamin E was associated with the occurrences of GDM (*p* < 0.001) and LGA (*p* = 0.004) in the second trimester (Table 3). Pregnant women who suffered from GDM had higher vitamin E concentrations in all trimesters than women without GDM. Women who gave birth to LGA infants had a higher serum vitamin E concentration in the first and second trimesters than women who gave birth to non-LGA infants (Table 4).

### 3.4. The Adjusted Association between Maternal Vitamin E Status/Concentration and Outcomes (GDM and LGA)

After controlling for potential confounders, multivariate analyses revealed that excessive vitamin E in the second trimester was a risk factor for the occurrence of GDM (OR = 1.640, 95% CI: 1.316–2.044) and LGA (OR = 1.334, 95% CI: 1.022–1.742) (Table 5). Restricted cubic spline analyses demonstrated that there were linear relationships between vitamin E and the outcomes (*p*-non-linear ≥ 0.05, Appendix A). Vitamin E concentrations in the first and second trimesters were positively associated with GDM (first: OR = 1.056, 95% CI: 1.038–1.073; second: OR = 1.062, 95% CI: 1.043–1.082) and LGA births (first: OR = 1.030, 95% CI: 1.009–1.051; second: OR = 1.040, 95% CI: 1.017–1.064) (Table 6).

## 4. Discussion

The present study found that maternal vitamin E levels increased from the first to the third trimester among our study participants. For mothers who had inadequate vitamin E levels, the majority had excessive levels in the second and third trimesters. After controlling for potential confounders, excessive vitamin E in the second trimester was a risk factor for GDM and LGA. Maternal vitamin E concentrations in the first and second trimesters were positively associated with the occurrence of GDM and LGA.

The maternal serum vitamin E levels in different trimesters demonstrated in our study were consistent with a number of studies in different areas of China, which reflected an increasing trend of vitamin E concentrations from the first to the third trimester [25,26,27,28,29] and a much larger proportion of excessive vitamin E cases than deficient cases [30,31,32]. For instance, a study among 12,340 pregnant women was recruited between 2013 and 2014 in the prenatal clinics of hospitals in seven districts (Haidian, Huairou, Dongcheng, Mentougou, Pinggu, Tongzhou, and Xicheng districts) of Beijing, China. That multidistrict study found that maternal serum vitamin E concentrations in the first, second and third trimesters were 9.10 ± 2.47, 14.24 ± 3.66 and 15.80 ± 5.01 mg/L, respectively. There were 5.6% women had abnormal vitamin E concentrations, and 5.37% had excessive levels [33]. Excessive vitamin E concentration during pregnancy may cause abortion and interfere with fetal development [6]. Owing to the vitamin E status of pregnant women in China, over-consumption or supplementation with vitamin E during pregnancy should be cautious.

The relationship between maternal vitamin E levels during pregnancy and GDM was not consistent in the literature. A meta-analysis of 11 studies reported that the level of vitamin E was significantly lower in the third trimester of pregnancy in GDM women in comparison to the healthy pregnant women [10]. However, the studies included were dominated by cross-sectional and case-control studies with small sample sizes (n < 100). In contrast, Song et al. selected 1000 pregnant women with GDM and 1000 pregnant women without GDM who were admitted to a hospital in Dalian, China from January 2017 to June 2018 and found that the serum vitamin E level of GDM pregnant women (21.34 ± 4.93 mg/mL) was significantly higher than that of non-GDM pregnant women (16.25 ± 5.49 mg/L) in late pregnancy [12]. Our results were similar to Song et al. [12], and we found that maternal vitamin E concentration in the second trimester was positively associated with the risk of GDM. Our results contributed to the literature in revealing the influence of maternal vitamin E during early and mid-pregnancy on the development of GDM. Further experimental studies to explore the related mechanism and prospective studies to confirm our results are warranted.

The association between maternal vitamin E and LGA in our study was consistent with a cohort study among 1231 pregnant women in the US. This US study measured maternal α-tocopherol levels at entry (16.0 ± 0.15 weeks) and at 28 weeks of gestation and found that each unit increase in α-tocopherol concentration increased the odds of delivering LGA babies by 8.8% at entry and 6.7% at week 28 [5]. Likewise, a prospective cohort study in South Korea found that birth weight was the highest when the concentrations of both vitamins C and E in the second trimester were high [34]. Besides, our result was similar to our previous exploration of the associations between maternal vitamin E and macrosomia [20]. The above evidence contributed vitamin E to be an antioxidant, which defensed against oxidative stress and impairment to fetal growth. Moreover, maternal GDM is a risk factor for LGA. It is likely that maternal vitamin E affects the development of GDM, and further influences the occurrence of LGA. It might be interesting to further explore the mediation or moderation role of GDM on the association between maternal vitamin E and LGA.

Our study has some strengths. First, unlike the majority of studies focusing on vitamin E status in a single trimester, our study explored the association between maternal vitamin E in every trimester and pregnancy outcomes. Changes in vitamin E levels from the first to the third trimester were also illustrated. Second, the sample size of this study was relatively large, and the potential confounders were adjusted in multivariate analyses. The accuracy of the results was thus enhanced. Limitations of our study should also be acknowledged. First, we were unable to collect blood samples in all trimesters for every participant; therefore, missing data in the exposure variables existed. Second, the generalizability of our findings was not confirmed because our study was conducted in a maternal and child hospital in a district of Beijing. Third, since it was a retrospective study using data from the hospital’s medical records, some potential confounders were not collected and analyzed, such as dietary intake and vitamin E supplementation. Finally, in our study, urine sample was not collected, and biomarkers of vitamin E were not examined. As a result, we were not able to study the metabolism of vitamin E [35] and understand the reasons for the excess of vitamin E among some present women. Further studies to address this aspect are warranted.

## 5. Conclusions

Vitamin E excess among pregnant women appears to be a public health issue in China. Our study demonstrated a positive association between excessive vitamin E in the second trimester and the risks of GDM and LGA. Further experimental studies are necessary to explore the mechanism between maternal vitamin E and the above pregnancy outcomes.

## Figures and Tables

**Table 1 nutrients-14-01629-t001:** Maternal characteristics and birth outcomes (n = 19,647).

	N	%
Maternal age		
≤20	137	0.7
21–30	12,752	64.9
>30	6758	34.4
Parity		
Primiparous	11,851	60.3
Multiparous	7796	39.7
Maternal ethnicity		
Han	18,472	94.0
Ethnic minorities	1174	6.0
Maternal education		
Below high school	4114	21.4
High school or college	7651	39.8
University or higher	7475	38.9
Maternal pre-pregnancy BMI		
<18.50	2123	11.0
18.5–23.99	12,144	63.1
24.00–27.99	3753	19.5
≥28.00	1236	6.4
Folic acid usage		
Yes	17,911	91.2
No	1736	8.8
Preterm birth		
Yes	771	3.9
No	18,827	96.1

**Table 2 nutrients-14-01629-t002:** Vitamin E levels during pregnancy.

	1st Trimester	2nd Trimester	3rd Trimester
	Median (IQR) or N (%)	Median (IQR) or N (%)	Median (IQR) or N (%)
Vitamin E	n = 16,705	n = 5520	n = 2190
Median, mg/L	10.00	14.60	16.00
Q1, Q3, mg/L	8.70, 11.50	12.30, 17.20	14.10, 18.40
Deficient	14 (0.1)	0 (0.0)	0 (0.0)
Adequate	16,650 (99.7)	5028 (91.1)	1892 (86.4)
Excessive	41 (0.2)	492 (8.9)	298 (13.6)

**Table 3 nutrients-14-01629-t003:** The association between vitamin E status and GDM and large for gestational age (LGA), by univariate analyses.

	Vitamin E Status	GDM	Non-GDM	*p* ^a^	LGA	Non LGA	*p* ^b^
		n (%)			n (%)		
First trimester	Not excessive	4026 (29.6)	9595 (70.4)	0.898	2814 (17.7)	13,100 (82.3)	0.333
	Excessive	10 (28.6)	25 (71.4)		9 (23.7)	29 (76.3)	
Second trimester	Not excessive	1095 (26.8)	2987 (73.2)	<0.001	887 (18.5)	3906 (81.5)	0.004
	Excessive	156 (39.5)	239 (60.5)		114 (23.9)	362 (76.1)	
Third trimester	Not excessive	330 (22.2)	1154 (77.8)	0.060	258 (14.7)	1494 (85.3)	0.415
	Excessive	66 (28.2)	168 (71.8)		46(16.6)	231 (83.4)	

^a^ The *p* value is reported from Chi-square test; ^b^ The *p* value is reported from Fisher’s exact test.

**Table 4 nutrients-14-01629-t004:** The association between vitamin E concentration (continuous variable) and GDM and large for gestational age (LGA), by univariate analyses.

	GDM	Non-GDM	*p* ^a^	LGA	Non LGA	*p* ^a^
	Median (IQR)			Median (IQR)		
First trimester	10.20 (8.90, 11.80)	9.90 (8.60, 11.30)	<0.001	10.30 (8.90, 11.80)	9.90 (8.70, 11.40)	<0.001
Second trimester	15.20 (13.00, 17.90)	14.30 (11.90, 16.80)	<0.001	15.20 (12.90, 17.70)	14.60 (12.30, 17.10)	<0.001
Third trimester	16.50 (14.50, 18.90)	15.95 (13.90, 18.20)	0.002	16.55 (14.45, 18.70)	16.00 (14.10, 18.40)	0.046

^a^ The *p* value is reported from Mann–Whitney U-test.

**Table 5 nutrients-14-01629-t005:** The association between vitamin E status and GDM and large for gestational age (LGA), by multivariate logistic regressions.

	GDM ^a^		LGA ^b^	
	OR (95% CI)	*p*	OR (95% CI)	*p*
First trimester vitamin E status (excessive)	0.691 (0.325–1.469)	0.336	1.312 (0.573–3.008)	0.521
Second trimester vitamin E status (excessive)	1.640 (1.316–2.044)	<0.001	1.334 (1.022–1.742)	0.034
Third trimester vitamin E status (excessive)	-	-	1.160 (0.773–1.742)	0.473

^a^ Maternal age, parity, maternal pre-pregnancy BMI, and folic acid usage were adjusted; ^b^ Maternal age, parity, maternal education, maternal pre-pregnancy BMI and gestational weight gain were adjusted.

**Table 6 nutrients-14-01629-t006:** The association between vitamin E concentration (continuous variable) and GDM and large for gestational age (LGA), by multivariate logistic regressions.

	GDM ^a^		LGA ^b^	
	OR (95% CI)	*p*	OR (95% CI)	*p*
First trimester vitamin E concentration	1.056 (1.038–1.073)	<0.001	1.030 (1.009–1.051)	0.005
Second trimester vitamin E concentration	1.062 (1.043–1.082)	<0.001	1.040 (1.017–1.064)	0.001
Third trimester vitamin E concentration	-	-	1.026 (0.983–1.071)	0.233

^a^ Maternal age, parity, maternal pre-pregnancy BMI, and folic acid usage were adjusted; ^b^ Maternal age, parity, maternal education, maternal pre-pregnancy BMI and gestational weight gain were adjusted.

## Data Availability

The data presented in this study are available on request from the corresponding author. The data are not publicly available due to privacy.

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
