# Peer review of "The Influence of Maternal Vitamin E Concentrations in Different Trimesters on Gestational Diabetes and Large-for-Gestational-Age: A Retrospective Study in China"

_nutrients, 2022, doi:10.3390/nu14081629_

Round 1

Reviewer 1 Report

The manuscript entitled „The influence of maternal vitamin E concentrations in different trimesters on pregnancy outcomes: a retrospective study in China” presents interesting issue, but some issues should be corrected.

Introduction:

Authors present a some basic and even very trivial information that should not be presented in a scientific manuscript, especially prepared for nutritional journal (e.g. “Vitamin E, a lipid soluble vitamin, is a family of eight natural isoforms, namely, α, β, γ, δ isoforms of tocopherol and α, β, γ, δ isoforms of tocotrienol”) – Authors should be aware that they do not prepare the basic manual for students, or column of the newspaper, but a scientific paper that should be interesting for researchers from the area of food and nutritional sciences, so they should understand that their readers will have the nutritional knowledge.

Authors should prepare this section not only to be interesting for Chinese/ Asian readers, but to be interesting for international readers. If Authors prepare their manuscript only for their national readers, they should publish it in some local journal. So, Authors should present here international data from various countries, not only the Chinese/ Asian ones.

Materials and Methods:

Information about informed consent provided by all participants should be indicated within this section.

It seems that Authors did not verify the normality of distribution of their data – they should do it and present the related methodology.

After verifying the normality of distribution, in case of parametric distribution mean ± SD should be presented, while for nonparametric distribution – median accompanied by minimum and maximum value.

The applied statistical test should be based on distribution

Results:

It seems that Authors did not verify the normality of distribution of their data – they should do it and present the related methodology.

After verifying the normality of distribution, in case of parametric distribution mean ± SD should be presented, while for nonparametric distribution – median accompanied by minimum and maximum value.

The applied statistical test should be based on distribution

Discussion:

Within the study there is no information about dietary intake or supplementation. If it was not studied – it should be indicated as a potential limitation of the study.

Conclusions:

This section is not the other abstract. Authors should only present direct conclusions from the conducted study (2-3 brief sentences)

Authors Contribution:

It seems that contribution of some Authors was only minor and they did not participate in preparing manuscript. There is a serious risk of a guest authorship procedure which is forbidden. In such case they should be rather presented in Acknowledgements Section and not be indicated as authors of the study.

Reviewer 2 Report

The authors in this paper reported the results of their study where they went to measure the levels of vitamin E in the serum of pregnant women on the assumption that vitamin E can protect pregnant women from oxidative stress and influence pregnancy outcomes. This study aimed to investigate maternal vitamin E concentration in each trimester and its associations with gestational diabetes (GDM) and gestational age (LGA). The authors used a very large cohort of pregnant women but doing a retrospective study with the aim of measuring serum levels of fat-soluble vitamins such as Vitamin E is not possible as this vitamin is subject to oxidation and the preservation of samples is essential to avoid analytical bias (see Torquato P. et al Free Radic Biol Med. 2021) .
 Given that the samples were collected from 2015 to 2018it would be important to know if the analyzes of Vitamin E serum levels were done at the time of collection in each quarter or done in subsequent years.
The authors noted that mean maternal vitamin E levels increased from the first to the third trimester. Furthermore, they saw that many mothers had an excess of Vitamin E and from correlation studies done they argue that an excessive level of Vitamin E in the second trimester was a risk factor for GDM and LGA. They observed that maternal vitamin E concentrations in the first and second trimester were positively associated with GDM and LGA.
The authors further and inappropriately state that avoiding excess vitamin E during pregnancy could be reducing GDM and LGA.

The study would have been interesting if the authors had gone to study the metabolism of this vitamin to understand the cause of the higher levels of this vitamin in some pregnant women (see Luo J et al. Nutrition, 2022; Bartolini D. et al, Antioxidant 2021; Pein H. et al., Nat Commun 2018; Giusepponi D et al., Talanta 2017 and others). The study of the serum and perhaps even urinary metabolome in these subjects would have allowed us to understand the cause of this excess in some women.

In my opinion, this paper has many methodological criticalities that need to be clarified in order to make certain statements, namely that excess vitamin E can cause gestational diabetes.

In details:
- the introduction is inadequate and the literature of pioneers in the study of this vitamin is not cited (as Traber MG, Niki E, Galli F, Birringer M, Cook-Mills JM, Eggersdorfer M, Frank J, Azzi A, Lorkowski S, etc.), only mostly Chinese authors are cited.
- as regards materials and methods, sampling must be better described, specifying when the analyzes were carried out. The standards used as a reference should be specified, the chromatographic column used, the extraction method of Vitamin E, etc.
- as far as the results are concerned they should also be shown as graphs with correlations, too many tables confuse the reader. Table 1 could go into supplementary data. If the serum level analyzes were done more than one month after sampling it would be appropriate to correct all results for the oxidized species levels of Vitamin E, tocopheryl quinone (alpha-TQ, see Torquato P et al., FRBM 2019 for analytic methods ), before making any correlations, elugubrations and claims.

Reviewer 3 Report

This study shows that pregnant women have higher vitamin E concentrations.

Comments:

  • While folic acid usage was documented, apparently the authors did not document the use of a multivitamin during pregnancy, which is the likeliest cause of an increase in vitamin E levels.  What other sources of vitamin E would be available?
  • The authors state that excess vitamin E can cause "maternal abortion" (I thought that only embryos or fetuses can be aborted) an "interfere with normal maternal development".  This is poor evidence, with just one reference.  What is meant by (ab)normal maternal development?
  • Vitamin E has been given as a preventive agent in women at high risk for pre-eclampsia, but again these data are not dealt with.
  • Introduction, page 2 line 47: "the incidence of GDM in China" depends on the screening and diagnostic methods.  These are not dealt with in the paper.
  • Introduction, page 2, line 58: "the prevalence of LGA ranged from ...".  Scientifically speaking, by definition, the 'prevalence' of LGA should equal 10% of the population, when local and updated birth weight charts are used.
  • Methods, page 2, line 89: "were incorrectly reported".  I presume that they were "missing"?
  • Methods, page 3, line 110: "Ethnic approval" is probably ethic.
  • Methods, page 3, lines 114-7: the methodology of the GDM label is insufficiently explained (glucose challenge? glucose tolerance?).  Again, the methodology of LGA label is insufficiently explained (local charts?).
  • Results: one would anticipate that other perinatal characteristics would be available apart from GDM and LGA, such as hypertension, pre-eclampsia, method of delivery, etc.
  • Discussion: once again, the potential 'risks' of 'excessive' vitamin E, if any, should be discussed.

Round 2

Reviewer 2 Report

The revisions turned out to be quite comprehensive. I only have a small fix to ask: the limitations of this study should be better emphasized and in Line 275-278 you would also add the references relating to the methods of measuring the metabolism of Vit. E including Bartolini D. et al Antioxidant 2021. it should also be noted  and inserted that this aspect relating to  Vitamin E metabolism will have to be taken into consideration in future studies

Author Response

Thank you very much for your valuable comments. The limitation section has been revised as suggested.

Line 275-279, “Finally, in our study, urine sample was not collected, and biomarkers of vitamin E were not examined. As a result, we were not able to study the metabolism of vitamin E [36] and understand the reasons for the excess of vitamin E among some present women. Further studies to address this aspect are warranted.”
